# What does patient feedback reveal about the NHS? A mixed methods study of comments posted to the NHS Choices online service

Gavin Brookes,[1] Paul Baker[2]

► Prepublication history forthis paper is available online.To view these files please visitthe journal online (http://dx.doi.org/10.1136/bmjopen-2016-013821).

## ABSTRACT

**Objective** To examine the key themes of positive and negative feedback in patients' online feedback on NHS (National Health Service) services in England and to understand the specific issues within these themes and how they drive positive and negative evaluation.

**Design** Computer-assisted quantitative and qualitative studies of 228 113 comments (28 971 142 words) of online feedback posted to the NHS Choices website. Comments containing the most frequent positive and negative evaluative words are qualitatively examined to determine the key drivers of positive and negative feedback.

**Participants** Contributors posting comments about the NHS between March 2013 and September 2015.

**Results** Overall, NHS services were evaluated positively approximately three times more often than negatively. The four key areas of focus were: treatment, communication, interpersonal skills and system/organisation. Treatment exhibited the highest proportion of positive evaluative comments (87%), followed by communication (77%), interpersonal skills (44%) and, finally, system/organisation (41%). Qualitative analysis revealed that reference to staff interpersonal skills featured prominently, even in comments relating to treatment and system/organisational issues. Positive feedback was elicited in cases of staff being caring, compassionate and knowing patients'' names, while rudeness, apathy and not listening were frequent drivers of negative feedback.

**Conclusions** Although technical competence constitutes an undoubtedly fundamental aspect of healthcare provision, staff members were much more likely to be evaluated both positively and negatively according to their interpersonal skills. Therefore, the findings reported in this study highlight the salience of such 'soft' skills to patients and emphasise the need for these to be focused upon and developed in staff training programmes, as well as ensuring that decisions around NHS funding do not result in demotivated and rushed staff. The findings also reveal a significant overlap between the four key themes in the ways that care is evaluated by patients.

[1]School of English, University of Nottingham, Nottingham, UK
[2]Department of Linguistics and English Language, Lancaster University, Lancaster, UK

**Correspondence to**
Dr Gavin Brookes;
gavin.brookes@nottingham.ac.uk

## Strengths and limitations of this study

► This study examines the largest (228 113 comments and approximately 29 million words) and most recent (March 2013 to September 2015) collection of online patient comments on NHS services analysed to date.

► Building on previous research, the feedback data examined relate to a wider range of areas of healthcare service provision, including: acute trusts, care organisations, care providers, clinical commissioning groups, clinics, dentists, general practitioner practices, hospitals, mental health trusts, opticians and pharmacies. Although the comments relating to these various areas of provision are not compared in the analysis, this nonetheless makes for a more widely representative dataset.

► The use of quantitative computer-assisted linguistic techniques produces large-scale, generalisable insights into this vast dataset, while more fine-grained, qualitative analysis helps elucidate nuances and areas of difference and overlap that have been overlooked by research employing solely quantitative approaches.

► Further data and research are required to assess possible demographic trends in the feedback given.

## INTRODUCTION

Since the 1980s, patient feedback exercises have been undertaken by an increasing number of healthcare providers worldwide in order to monitor the quality of the services they provide and stimulate improvements where needed.[1] Although the reliability of patient feedback as an indicator of the technical quality of care remains a topic of debate,[2] patient feedback exercises have nonetheless become a staple way of measuring and regulating healthcare standards,[3][4] as well as ensuring public involvement in the design and improvement of healthcare provision.[5] Patient empowerment is, as Gann puts it, 'here to stay' (p. 150),[6] and policy makers over the world have come to recognise the potential of active patient involvement to drive service improvements, improve self-care and ultimately improve the affordability and

sustainability of the services they provide.[7] In England, since 2002, patient feedback has played an increasingly significant role in the way that care quality is assessed, with all National Health Service (NHS) trusts required to collect and report the results of feedback on their services to the regulatory body, the Healthcare Commission. The importance of the insights gained from patient feedback exercises is all the more pronounced in this context, where reductions in government expenditure in areas of social provision have required healthcare providers to constantly demonstrate both the quality and financial viability of the services they provide.

Healthcare providers can obtain feedback from their patients using a range of methods which can be implemented in different settings and at differing times following an episode of treatment. Ziebland and Coulter[8] provide a list of such methods, which include (but are not limited to): face-to-face interviews, postal questionnaires, telephone interviewers (using automated and live interviewers), web-based online questionnaires, diaries, questions on handheld portable devices, touch screen kiosks and bedside consoles. Moreover, feedback can be collected on-site, at the point of service contact or at patients' homes, some days, weeks or months later.[8] The analysis reported in this study focuses on feedback given in the form of online patient comments. In recent years, increasing attention has been paid by researchers and healthcare providers to the internet as a site for patients to recount their express of healthcare services and to draw attention to what was good and bad about those experiences.[9–11] One such recent study was undertaken by Greaves *et al*,[12] who compared patients' ratings of care posted to the NHS Choices online service with the results of non-experiential measures of service performance, such as morality rates. The researchers reported that, overall, patients' ratings tended to correlate with the non-experiential measures. For example, hospitals that were poorly evaluated by patients were found to have higher mortality rates. This research therefore supports the value of online forms of patient feedback for assessing care quality and targeting areas for improvement.

Given the increasing significance of patient feedback to the ways that healthcare services are designed, delivered and regulated, there is a pressing need for research that accounts for the concerns expressed by patients in their feedback. However, rather than explore the content of patient feedback itself, the majority of existing research in this area is concerned chiefly with: reviewing the suitability of instruments and methods of collecting and analysing feedback[13]; considering the reliability of feedback data for assessing healthcare quality[14]; reflecting on the extent to which insights gained from such exercises have actually improved service provision[15] and recommending how such insights might be translated into positive clinical outcomes in the future.[16] The comparatively few studies that have examined the content of patient feedback (even fewer of which relate to healthcare in England) have reported recurring drivers of feedback to include the technical quality of care, accessibility to care, and the interpersonal and communication skills of practitioners (with the latter two often conflated).[17–19]

The present study identifies and examines the key drivers of positive and negative feedback on healthcare services given in patients' online comments posted to the NHS Choices website between March 2013 and September 2015. Our findings build on existing patient feedback research in several important ways. At 228 113 comments and approximately 29 million words, the feedback data we analyse are considerably larger than those examined in previous research on this topic, which have mainly accounted for hundreds of comments[20–22] and at the most in the tens of thousands.[23] Moreover, the data we analyse represent feedback relating to a wider range of healthcare services than considered in previous research, which has often focused on specific areas of healthcare provision.[24] The lion's share of research on patient feedback was conducted using data collected in the 1990s and early 2000s, while our dataset contains comments made as recently as September 2015, making this dataset the most up-to-date of its kind. More broadly, given the ever changing landscape of healthcare provision in England and the UK, the present study responds to the need for regular and up-to-date research that assesses patient attitudes towards the NHS and specifically identifies the key drivers of feedback about the particular services they access.

## METHODS

### Data

We studied the written feedback posted to the NHS Choices online service (http://www.nhs.uk/pages/home.aspx) between March 2013 and September 2015 (data made available to the researchers). The comments were collected from the NHS Choices service's comprehensive RSS feed for posted comments, using a developer key provided to us for this purpose, and then converted from RSS/XML to suitable structured corpus/database format for analysis. The data comprise a total of 228 113 comments, amounting to 28 971 142 words. The comments relate to a variety of healthcare organisations, including acute trusts, care organisations, care providers, clinical commissioning groups, clinics, dentists, general practitioner (GP) practices, hospitals, mental health trusts, opticians and pharmacies. However, the majority of the comments (27 005 715; 93.21%) relate to three primary care services: GP practices, hospitals and dentists. A numerical breakdown of the data is provided in table 1.

### Analysis

We examined the comments using computer-assisted methods of linguistic analysis afforded by CQPweb,[25] an online tool that offers a range of techniques for quantitatively and qualitatively analysing large collections of digitised language data. We began by identifying the 10 most frequently occurring linguistic markers of positive and negative evaluation across the comments. These

| Table 1 | Breakdown of the NHS comments database | | |
| --- | --- | --- | --- |
| Section | Comments, N | Word counts | Mean words per comment |
| Acute trusts | 1 022 | 159 385 | 156 |
| Care organisations | 6 | 1 164 | 194 |
| Care providers | 4 493 | 422 133 | 94 |
| CCGs | 1 | 253 | 253 |
| Clinics | 2 887 | 400 813 | 139 |
| Dentists | 41 958 | 4 306 698 | 103 |
| GP practices | 111 318 | 14 093 437 | 127 |
| Hospitals | 55 145 | 8 605 580 | 157 |
| Mental health trusts | 565 | 111 557 | 197 |
| Opticians | 1 734 | 179 493 | 104 |
| Pharmacies | 8 984 | 690 629 | 77 |
| All sections | 228 113 | 28 971 142 | 146 |

GP, general practitioner; NHS, National Health Service; CCGs, Clinical Commissioning Groups.

words were manually identified from a list of all the words occurring in the data provided by the 'frequency' function of CQPweb. Evaluation is a complex linguistic phenomenon, and can be made according to a variety of parameters, including the extent to which things are important, expected, comprehensible, possible and reliable. To ensure that our analysis captured the broadest range of themes concerning the positive and negative evaluation in the comments, we focused on the most generic evaluative items, that is, words that were broadly used to describe something as either being good or bad.

Using CQPweb, we then generated a list of those words that tend to occur frequently alongside the positive and negative evaluative words in the comments, that is, their most frequent 'collocates'. Collocation refers to 'the characteristic co-occurrence patterns of words'.[26] By analysing the collocates of the evaluative words, we were able to get a sense of what tended to be the target of the evaluation in the feedback—that is, of what was evaluated as 'good' and 'bad' in the comments. These words therefore reflect the key themes of positive and negative feedback in the data.

Building on this, the next more qualitative step in our procedure involved closely reading a randomly selected sample of comments in which each theme was evaluated positively and negatively to determine the more specific reasons or 'drivers' of the evaluation. Each sample consisted of 100 comments and contained comments relating to all organisations represented in the data. To ensure that 100 comments provided a sufficiently representative sample for this stage in our analysis, we adopted a saturation point procedure, well established in such quantitative linguistic research,[27] of randomly selecting 30 comments, analysing the emergent patterns, proceeding to examine another 30 randomly selected comments and continuing the process until saturation point was reached and new patterns had ceased to emerge. New patterns, or drivers, were no longer emergent by the time we analysed the 100th comment (positive and negative) for each theme, and so this sample size was deemed sufficiently large to account for the common drivers of positive and negative feedback, yet small enough to facilitate fine-grain qualitative examination.

## RESULTS

### Quantitative findings

Table 2 displays the 10 most frequent positive and negative evaluative words used in the comments. The comparatively higher frequencies of the positive words (total: 223 439) compared with the negative words (total: 73 363) provide a quantitative indication that the patients are more likely to evaluate the services they access positively than negatively. The positive evaluation words occur,

| Table 2 | Ten most frequent positive and negative evaluative words in the comments | | | | |
| --- | --- | --- | --- | --- | --- |
| Positive | | | Negative | | |
| Word | Frequency | Comments, N | Word | Frequency | Comments, N |
| Good | 59 237 | 46 192 | Bad | 16 945 | 14 798 |
| Excellent | 49 090 | 38 128 | Poor | 15 274 | 12 548 |
| Great | 34 298 | 27 772 | Worst | 7 627 | 6 627 |
| Best | 25 556 | 21 641 | Worse | 7 289 | 6 447 |
| Fantastic | 15 186 | 12 915 | Terrible | 6 799 | 5 920 |
| Brilliant | 11 546 | 10 136 | Awful | 6 106 | 5 291 |
| Wonderful | 10 371 | 8846 | Appalling | 4 410 | 4 007 |
| Amazing | 9749 | 8081 | Disgusting | 3 246 | 2 913 |
| Outstanding | 5019 | 4277 | Ridiculous | 3 206 | 3 023 |
| Exceptional | 3387 | 3060 | Useless | 2 461 | 2 217 |
| Total | 223 439 | 18 105 (mean) | Total | 73 363 | 6 379 (mean) |

on average, across almost three times as many comments as the negative words.

CQPweb was then used to generate lists of those words occurring frequently within the three words preceding and following the positive and negative evaluation words in table 2 (ie, their collocates). The collocational span of 3 is the default value in CQPweb and is fairly standard in collocation analyses, which tend to operate with spans ranging from three to five words.[28] As we mentioned earlier, evaluation is a complex linguistic phenomenon, and the positive words below could, in some circumstances, be used to evaluate something negatively ('not good') and vice versa (eg, 'isn't bad'). Such cases comprised a tiny proportion (under 1%) of cases and were removed from the remainder of the analysis below.

The 100 most frequent words occurring alongside the positive and negative words were thematically coded to reflect the most frequently evaluated areas of concern for patients giving feedback. Four areas emerged as frequent across the comments (corresponding words in brackets): (1) treatment (*care, treatment, dental*); (2) communication (*communication, attention, listener(s), advice*); (3) interpersonal skills (*atmosphere, attitude(s), manner(s)*) and (4) system/organisation (*system, appointment, management, waiting time(s)*). As the forthcoming qualitative analysis shall demonstrate, feedback concerning communication and interpersonal skills related to a mixture of medical and non-medical staff groups, with the latter including staff members such as receptionists and managers. Note that we combined *waiting* and *time(s)* together into one linguistic item, as references to *time(s)* by itself often appeared in statements like 'I had a really bad time', which were too vague to be categorised. Based on the corresponding words (in brackets), we then examined how often each

concern featured alongside the positive versus negative evaluative words (figure 1).

Of the four key themes displayed above, treatment exhibited the highest proportion of positive feedback, occurring alongside the positive evaluation words 87% of the time. Communication was also evaluated positively overall (77%), while interpersonal skills were only evaluated positively 44% of the time and system/organisational issues fared worst of all with only a 41% positive evaluation.

### Qualitative findings

To understand why the four key themes identified in our quantitative analysis were evaluated positively and negatively across the patients' comments, we examined a sample of comments (n=100) in which each occurred alongside the positive and then negative evaluation words. Tables 3–10 report the reasons each theme was positively and negatively evaluated in our sample. This section deals with each theme in turn, starting with the theme that fared best in the patients' comments (treatment), and concluding with the theme that fared worst (system/organisation).

Treatment

Communication

Interpersonal skills

System and organisation

## DISCUSSION
### Statement of principal findings

The online comments analysed in this study paint a generally positive picture of healthcare services provided by the

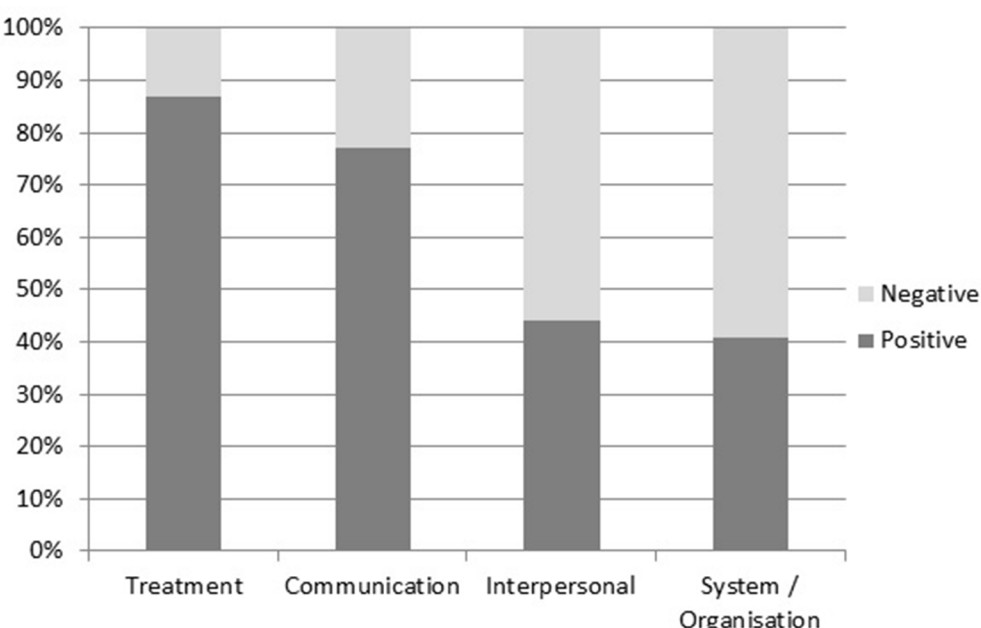

**Figure 1** Collocation of most frequent feedback themes with positive and negative evaluation words.

**Table 3** Reasons treatment was positively evaluated

| Category | Percentage of sample (%) | Example of comment |
|---|---|---|
| Good interpersonal skills | 47 | Excellent Dental Care and service, Highly recommend I have been a patient at [name] for approx 8 years, and thought I would drop a line to say how satisfied I am with the full service the practice provides. The staff are friendly, polite and professional and provide sufficient information at all times. |
| Good communication skills | 16 | Always great treatment, reception staff are also very good at explaining all options. Cannot recommend the team highly enough. |
| Technically competent | 10 | Excellent customer care Excellent service from a friendly knowledgeable team. |
| Patient centred | 9 | The dental care is excellent treating one with respect and discussing with you treatment options. I would particularly recommend the senior nurse who provides a very individualised care. |
| Efficient | 8 | Good Surgery I am a member of the health service and I have to say that everyone at the surgery is very helpful and friendly and the treatment I have received has been very efficient and effective in dealing with my concerns. |
| Hardworking | 5 | I don't think the people who have made negative comments understand how difficult it can be for a GP practice to try and meet the demands of today's patients, what you have here is a committed team of GPs, nurses and admin staff who are working hard daily to give patients their best service and care. |
| Clean facilities | 3 | I received wonderful care during my 6 weeks there and it was very clean too. |
| Good food | 2 | After my surgery was done the nurses took great care of me and the food was 1st rate. |

NHS in England. Our quantitative analysis of the patient comments revealed the most commonly used linguistic markers of positive evaluation occurred approximately three times as often as markers of negative evaluation and, on average, across approximately three times as many comments. Patients' experiences and impressions of their treatment, communication, staff members' interpersonal skills and system/organisational issues were identified as

**Table 4** Reasons treatment was negatively evaluated

| Category | Percentage of sample (%) | Example of comment |
|---|---|---|
| Poor interpersonal skills | 35 | Nursing staff are insensitive, they have poor insight in how to care for the mentally ill. They are bullies and self-centred. Patients are picked on and have the micky taken out of them by nursing staff. An absolute shambles and one of the worse care providers. |
| Lack of appointment availability | 25 | Not only can you never get an appointment, but the extremely rude and unhelpful receptionist tops this GP surgery to be surely one of the worst for care of patients. |
| Technically incompetent | 18 | Ward 23 Appalling care. Our relative was admitted for simple antibiotics for an infection in her leg. 2 days later she is bed bound after having her other medical complications completely ignored. |
| Poor communication skills | 8 | Simple nursing care non-existent, communication to family again non-existent. |
| Lack of aftercare | 6 | Aftercare seems to be something they have no idea exists… instead of being a standard procedure it is totally ignored. That tells me they do not care. Absolutely Awful - If you can go to a different GP then do! |
| Unclean facilities | 4 | The place is dirty and the standard of care is so bad I honestly can't believe the hospital has not been shut down. |
| Difficulty accessing test results | 2 | Poor customer care. The GPs themselves are good. However, I had blood tests done recently and have had a nightmare trying to find out the results. |
| Lack of seating and space in waiting areas | 2 | People that were visibly ill were holding themselves up against the walls and sitting on the floor. It was absolutely shocking. Probably the worst care I have ever seen within the NHS. Again, no further seating was provided. |

**Table 5** Reasons treatment was positively evaluated

| Category | Percentage of sample (%) | Example of comment |
|---|---|---|
| Listens to patients | 44 | He is an extremely skilled dentist who pays great attention to detail and to listening to his patients… concerns. |
| Good interpersonal skills | 29 | The most best gp I have ever met, so nice to talk, very patience dr, more professional, very good listener very helpful, very kind! |
| Good communication between staff members | 8 | Excellent Team The [NAME] Practice offers an excellent team approach to healthcare, as far as I can see all services are fully integrated with excellent communication between all disciplines. |
| Not rushed | 7 | I have found my doctor to be a very good listener and though doctors are very busy, I have never felt my consultations rushed. |
| Sensitive | 5 | Communication was excellent, given in a sensitive caring manner and straightforward clarity. |
| Frequent | 3 | Communication wad excellent and I was always kept up to date by my surgeon. |
| Involves patient's family | 3 | The communication was excellent with my husband being able to ask questions about the timing of the surgery, what to expect & how he could help in my recovery! |
| Perceived as honest | 1 | This to me is the best NHS GP surgery in the area! Caring and second to none treatment honest good advice. |

**Table 6** Reasons communication was negatively evaluated

| Category | Percentage of sample (%) | Example of comment |
|---|---|---|
| Does not listen to patients | 32 | Patronising and a bad listener I have decided to register with another GP after being a patient at this clinic for 4 year. I feel that one doctor doesn't really take the time to listen, even if it is a problem that may be sensitive or embarrassing. |
| Poor interpersonal skills | 29 | The two doctors have bad attitude and very bad advice and clearly don't care about your welfare. Totally avoid. |
| Poor interstaff communication | 14 | There is poor communication between admin here and medical staff. Results go into a black hole and don't seem to reach or be reviewed or followed up by doctors. |
| Failure to make contact to arrange appointments | 9 | We waited 6 weeks to receive an appointment for our son despite continued phone calls to the consultants secretary and a business manager and various admin staff. NTW obviously feel at this level of communication is adequate. When I did ring on numerous occasions I was assured a response but not rang back. |
| Unreliable telephone services | 7 | My complaint is in relation to poor communication from outside agencies and from myself trying to access support via telephone. There is an online service but that advises you not to leave anything important as it is not always checked. |
| Failure to involve patients' relatives | 4 | However after another phone call to the A & E sister she explained very clearly that my mother had in actual fact gone from them to the [NAME] ward at around 04.30 and they did not want to contact me at that early time but expected the ward staff to do this after admission to their ward. The Sister agreed that it was poor communication, especially as my mother is very elderly and frail and potentially having fractured her hip she may have died. |
| Rushed | 4 | Rushed consultations and poor communication. |
| Poor English language skills | 1 | Front line staff are poor in communication and poorly trained except for the two senior staff. Some of the staff at the reception are unable to express (in English) causing miscommunication and stress to patients. |

**Table 7** Reasons interpersonal skills were positively evaluated

| Category | Percentage of sample (%) | Example of comment |
|---|---|---|
| Friendly and approachable | 41 | Some emergency Dentists treat people with indifference but not this one, lovely warm people, great atmosphere, many thanks. |
| Good communication with patients | 27 | The doctor I saw had an excellent manner. Answered all my questions. Wonderful service. |
| Listens to patients' concerns | 11 | On the positive, the doctor has a great manner with their patients and takes time to listen to your story and symptoms. |
| Empathetic | 9 | They were brilliant and really cared; they had a fantastic bedside manner and is genuinely empathetic. |
| Patient centred | 7 | The doctors 'bedside manner' is great. They take time to listen to concerns and talk through clearly and carefully the options to deal with my issues and problems. |
| Smiles | 3 | The nursing staff were very friendly and helpful, and with a very good attitude, always smiling which I imagine would go a long way to helping relieve any nerves some people may have. |
| Good sense of humour | 2 | Can't fault the practice, courteous service good communication, all done with a relaxed manner and good humour. |

key to the ways that healthcare services were both positively and negatively evaluated. Of these key themes, treatment exhibited the highest proportion of positive feedback, occurring alongside the positive evaluation words 87% of the time. Communication was also evaluated positively overall (77%), while interpersonal skills were only evaluated positively 44% of the time and system/organisational issues fared worst of all with only a 41% positive evaluation.

Qualitative examination of the data was able to reveal the more precise nature of the evaluation made in the comments, as well as uncover overlaps and nuances between the key themes of positive and negative feedback. This part of the analysis suggested that as well as constituting a key theme in its own right, staff members' interpersonal skills also emerged as a frequent driver of both positive and negative feedback in relation to how treatment and staff communication skills were evaluated. Other frequently cited drivers of positive and negative feedback included accessibility to care, patient centredness, and staff-to-staff and staff-to-patient communication. Staff technical competence was a less prominent driver of feedback, cited only in relation to the evaluation of treatment itself, and accounting for a relative minority of these comments. Our findings therefore support the notion that there is a discord between the significance that practitioners and patients place on technical competence when judging the overall quality of care.[29]

### Strengths and weaknesses of the study

This study has examined the largest and most up-to-date collection of patient feedback on NHS services in England in any format. Our use of quantitative computer-assisted linguistic techniques has produced large-scale,

**Table 8** Reasons interpersonal skills were negatively evaluated

| Category | Percentage of sample (%) | Example of comment |
|---|---|---|
| Rude/impolite | 54 | Receptionist staff are disrespectful and have a complete bad attitude towards me and others over the phone, some may be kind but other than that they have a disgusting attitude, as for doctors, they are ok since they listen to me. |
| No one answers the phone | 16 | Extreme bad manners when you have to call the surgery 8 times (and counting) to ask to speak to a doctor (during the times they recommend). |
| Dismissive | 11% | Bad manners Reception have a terrible attitude, very rude and dismissive. |
| Lazy | 11 | Worse still when I try and book a later date the only GP that is available is awful. Their attitude is diabolical and the snorting all the time is so off putting. I find them so rude and their lethargic attitude annoys me. |
| Does not listen to patients | 5 | Never have I witnessed such bad manners from a healthcare professional. Did not listen or care about anything I had to say. |
| Not smiling/look unhappy | 3 | One of the drs for children is especially bad attitude, they always showing unhappy face to my children who is afraid, and they were rude to my partner one time when he was not speaking good English. |

**Table 9**  Reasons system and organisational issues were positively evaluated

| Category | Percentage of sample (%) | Example of comment |
|---|---|---|
| Good appointment availability | 30 | Excellent appointment availability I understand this surgery is a training practice which is supervised by permanent senior doctors. |
| Good appointment booking system | 25 | The online appointment system is excellent and takes pressure off phone lines and means that it is easier to find an appointment that suits, with a preferred GP If need be. |
| Short waiting times | 23 | Very pleased [name] ward - Great hygiene, friendly members of staff, good waiting times, reassuring and good with patients. |
| Good telephone services | 11 | The doctor call back system is an excellent use of resources and is a good triage system ensuring that a patient is seen by the appropriate person. |
| Good availability of emergency appointments | 8 | I have always been able to get an appointment in an emergency in good time. |
| Good prescription system | 3 | I use the new on line system which is excellent for requesting repeat prescriptions |

generalisable insights into this dataset. Yet at the same time, the more fine-grained, qualitative analysis was able to elucidate areas of difference and overlap that have been overlooked by research employing solely quantitative approaches in the past.

Although this dataset has proven to be a valuable resource for learning about individuals' perspectives on the healthcare services they access, its lack of metadata regarding the demographic information of individual contributors meant that it was not possible to attribute particular types of comment or concern to any demographic group. It is also worth bearing in mind that the majority of the comments we analysed (93.21%) relate to the primary care services of GPs, hospitals and dentists.

While this is unlikely to present issues respecting the general trends examined, more specific comments relating to these areas might be said to be over-represented compared with other areas of service provision, such as care providers and mental health trusts. Moreover, the data analysed in this study represent feedback given in one specific form (online comments), posted to one particular website (NHS Choices), about organisations based in one country (England). This raises issues surrounding representativeness; for those who choose to share their experiences online are not necessarily representative of the general population. It is now well documented that, compared with non-internet users, internet users tend to be younger, are more educated

**Table 10**  Reasons system and organisational issues were negatively evaluated

| Category | Percentage of sample (%) | Example of comment |
|---|---|---|
| Poor appointment availability | 27 | Poor appointment availability I tried to make an appointment for four days ahead but was told all appointments on that day were embargoed that is, not available, and so I was advised to phone back next day. |
| Long waiting times | 22 | Waiting time awful I had a scheduled appointment and waited for more than 1.5 hours and still I was not seen by the doctor, nor was I given any explanation or information as to when I would be seen. |
| Poor appointment booking system | 19 | The appointment system very bad and the phone is constantly engaged and when you do ring for an appointment there is a silly system there that only takes 3 days advance booking. |
| Long telephone waiting times | 17 | The telephone system is awful, I am told to press zero to speak to someone then told to ring back later by an automated voice because they are busy. This costs me money to ring and is very frustrating. |
| Poor appointment time management | 10 | Terrible long waiting time for appointments Every time I go in to see the GP, I have to wait at least 40mins after my appointment time to see a doctor, it's really annoying that time is not managed effectively here. |
| Limited opening hours | 3 | Wake up Mere Lane modernise your service to the standard of your Building & Doctors Great doctors, great facilities, awful appointment service & opening hours! |
| Lack of availability of emergency appointments | 2 | Staff are very rude and not interested in your wellbeing, telephone appointments are at a far reach as its always fully booked, no surprise there as there opening hours are awful. Even for an emergency appointment. |

and are from higher income brackets.[30] Although this digital divide is estimated to have narrowed over time,[11] the perspectives of people from these so-called 'hard-to-reach' or 'seldom-heard' groups are still likely to be under-represented in our data.[8 31]

### Strengths and weaknesses in relation to other studies and important differences in results

Where a great deal of existing research has explored patient feedback in terms of predetermined themes, the data-driven approach adopted in the present study has allowed drivers of feedback to emerge from the comments themselves throughout the course of the analysis. As a consequence, system and organisation issues, which have remained largely unexplored in existing research, have emerged here as significant drivers of positive and negative evaluation with respect to various other aspects of care, including quality of treatment and staff communication skills.

As well as providing fresh insight into the perspectives of patients accessing contemporary healthcare services in England, the findings reported in this study also provide more substantive quantitative evidence to support the findings reported in existing studies of patient feedback that are based on comparatively smaller and older datasets.[32 33] However, our findings highlight the centrality of interpersonal skills as a key area of concern in its own right and as significant to the ways that treatment quality and staff communication are evaluated.

### Meaning of the study: possible explanations and implications for clinicians and policy makers

While the majority of research into patient feedback has focused principally—in most cases exclusively—on what motivates negative feedback, the present study has elucidated the drivers of positive and negative feedback equally. Accordingly, while the reported drivers of negative feedback might flag up areas that require attention, the specific drivers of positive feedback outlined over the course of the analysis offer insight that can be used to stimulate and guide quality improvement efforts.[34–36]

The quantitative section of our analysis suggested system and organisational issues to be a prominent theme in negative feedback. This often relates to issues surrounding accessibility of care, such as (emergency) appointment availability, waiting times, technical difficulties experienced with online booking systems, telephone waiting times and practice opening times. Tightening government expenditure in healthcare provision and resultant constraints on practitioner time and availability mean that these issues are unlikely to abate. Such issues arguably lie within the remit of policy makers and governing bodies. However, practitioners and other staff can improve patient feedback in this area by making an effort to ensure that appointments run on time and informing and updating patients and their families in the case of cancellations or delays.

The qualitative section of our analysis suggests that staff interpersonal skills lie central to improving care, as these were shown to motivate negative (and positive) feedback in relation to a variety of areas of concern. On the surface, the findings of this study suggest that developing the interpersonal skills of staff should be a priority in staff training. The interpersonal skills of both medical and non-medical staff were evaluated positively for qualities such as being friendly and approachable, empathetic, for smiling, and not being afraid to laugh and joke with patients. Allied to this, practitioners were frequently positively evaluated for providing care that was patient centred and involved discussing treatment options with patients, as well as explaining treatment plans and listening to their concerns. Conversely, staff were negatively evaluated when they were perceived as being rude, dismissive, lazy, not listening to patients' concerns, as well as for not smiling and appearing unhappy.

Most professional medical training (eg, in medicine and nursing) includes the development of communication skills as a key element at the undergraduate level and onwards. This kind of interpersonal training is often focused on developing skills such as information gathering and shared decision making. Our findings suggest that, as far as patients are concerned, the interpersonal aspect of interaction is given a high premium. Such skills might be developed more effectively through greater opportunity for hands-on human engagement, rather than instruction alone, at the early stages of training. In terms of developing the interpersonal skills of other non-medical staff groups, it is likely that many staff working in administrative capacities, such as receptionists, will not have received formal training in interpersonal skills (although healthcare providers are increasingly running courses in 'customer service' to address this). Likewise, there is an intermediate group, which includes staff working in healthcare assistance, who may have received limited or no formal training, but who nonetheless engage with patients at very significant levels and may benefit from some form of interpersonal skills training.

However, many of these specific interpersonal or 'soft' skills can be linked to the concept of emotional labour,[37] which involves the regulation of emotion to create a publicly visible facial and bodily display within the workplace. These kinds of attributes might seem more like individual character traits, and so incorporating these into training poses challenges, especially as members of the public are often unimpressed by 'scripted' interactions which are rightly seen as inauthentic.[38] Soft skills training, while clearly helpful in some areas, may sometimes be a 'sticking plaster' solution to cover for wider structural problems involving overstretched systems. Other positively evaluated interpersonal aspects of care, such as involving patients in communication and decision making and ensuring that patients have sufficient time to interact with medical staff and are not made to feel as though they are 'rushed,' might constitute more tangible and attainable targets for skills development programmes.

In an effort to stimulate such improvements, the findings from the project from which this research derives, including the results reported in this article, have been presented to the Insight and Feedback Team at NHS England as well as the Care Quality Commission in the UK. However, in reality, translating such findings into practice is seldom straightforward. After all, although most major public sector healthcare providers collect feedback on their services at least annually, this information is not always used to improve service quality.[3] Suggested reasons for this include a lack of attention to patients' experience at senior levels,[39] as well as feedback data not being specific to particular wards or teams.[40]

As well as gesturing towards areas for improvement in healthcare provision, the findings reported here also provide insights into patient feedback more generally; insights that likely bear implications for how such feedback should be interpreted in the future. Our quantitative examination of the data revealed significant overlap between the drivers of positive and negative feedback. As an example, although treatment fared best of these four themes of feedback in terms of positive evaluation, it was only by examining the comments relating to treatment that we were able to show that 47% of the positive comments relating to this theme actually praised interpersonal aspects of care, rather than the technical competence of staff, which accounted for only 10% of these comments. It is therefore beneficial, where possible, to gauge feedback at a granular level. This is where combining quantitative and qualitative approaches can bear significant advantages for researchers, allowing us to deal with large datasets in a way that is sensitive to subtle nuances and overlaps and to point to specific areas for praise or improvement that only become apparent at the more granular level.

### Unanswered questions and future research

Although the data contain feedback relating to a variety of healthcare organisations, such distinctions have not figured in the analysis undertaken in this study. Future studies on this particular dataset should therefore take a modular, even comparative approach, to ascertain similarities or differences in the ways that care is evaluated in each of these areas. Furthermore, we did not have access to the demographic information of the comment posters. Future research should endeavour to collect and examine data that comprise this kind of demographic metadata in order to determine whether particular concerns are attributable to people living in certain locations or belonging to particular age, ethnic or sex-related groups. Future research should also assess feedback given in other, particularly non-digital, mediums in order to help account for the perspectives of patients from 'hard-to-reach', 'seldom-heard' groups who might be less likely to give feedback online. Ziebland and Coulter[8] recommend that non-traditional methods of data collection, such as pictures, stories and drama, might be used to incorporate the views of such groups in future feedback collection exercises.

**Acknowledgements** We wish to thank Dick Churchill for his advice concerning the pedagogical implications of our findings.

**Contributors** GB co-planned and co-conducted the research, and took charge of writing the paper. He is responsible for the content as guarantor. PB co-planned and co-conducted the research and contributed to the writing of the paper. Both authors had access to the data.

**Funding** Economic and Social Research Council; grant number: ES/K002155/1.

**Disclaimer** All authors have completed the ICMJE uniform disclosure form and declare: all authors had financial support from the Economic and Social Research Council for the submitted work; no financial relationships with any organisations that might have an interest in the submitted work in the previous 3 years; no other relationships or activities that could appear to have influenced the submitted work.

**Competing interests** None declared.

**Ethics approval** Research ethics approval for the study was obtained from Lancaster University, Lancaster, UK.

**Provenance and peer review** Not commissioned; externally peer reviewed.

**Data sharing statement** No additional data available.

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
