## [Reviewer comments · BMJ Open]

ARTICLE DETAILS

TITLE (PROVISIONAL)	What does patient feedback reveal about the NHS? A mixed methods study of comments posted to the NHS Choices Online service
AUTHORS	Brookes, Gavin; Baker, Paul

VERSION 1 - REVIEW

REVIEWER	Anne-Marie Boylan University of Oxford, United Kingdom I conduct research on this same topic.
REVIEW RETURNED	24-Oct-2016

GENERAL COMMENTS	This paper addresses the highly relevant, topical and important issue of analysing online patient feedback. It employs a novel method to explore a huge amount of patient comments and, to date, is the only study to attempt something of this magnitude in relation to online reviews of the NHS. However, the majority of the data comes from primary care and I wonder if the study would be more usefully conceptualised as a study of primary care rather than the NHS in general. As the paper stands, more information is needed in the methods section to fully evaluate if the results address the objective. Please explain how you accessed the data from NHS Choices; the process followed to identify the ten most frequently occurring linguistic markers of positive and negative evaluation; and how you sampled 100 messages for qualitative analysis. Given the size of your dataset, how did you ascertain that a qualitative examination of 100 comments would be sufficient? Did you aim for or reach saturation? Did you sample comments on all services? Please also explain the qualitative analytic process you followed to analyse the data, and supplement your qualitative findings with examples of the comments. In the results section, table 2 shows the 10 most frequent positive and negative evaluative words. How can you be sure that the words were always used in a positive or negative way in the comments? For instance, the words good or great could have been negatively used as in 'not great' or 'not good'? Or the words bad or worst could have been used in comments as 'not bad' or 'not the worst'. I'm sure you've considered this, but please reflect on this in the paper. One of the key findings is that comments frequently mentioned staff interpersonal skills. It would be helpful to know who is defined as 'staff' (this may well include GP receptionists, for instance) as this inevitably has implications for any recommendations based on your
---

	findings. Finally, there is a growing body of research on online patient comments that has not been cited here. It is clear the authors are aware of this, but it would be helpful to position this study in relation to work conducted elsewhere.
--	--

REVIEWER	Glenn Robert King' College London, UK
REVIEW RETURNED	09-Nov-2016

GENERAL COMMENTS	This is an interesting paper providing a description of a relatively novel form of computer-assisted analysis of a large dataset relating to patients feedback on their experiences of NHS care. Throughout I felt the paper needed greater attention to the specific details of several aspects of the study and to draw more explicitly on the wider literature relating to gathering and using patient feedback in healthcare systems and organisation (which has highlighted the relational/transactional nature of staff/patient interactions which forms the main finding here). I have made several suggestions in this regard below. Throughout the manuscript it would be helpful if the authors could make clear to the reader that their study is focused on just one specific form of patient feedback (online comments and ratings) to one particular website (NHS Choices). In various places the phrase 'patient feedback' (e.g. first sentence, 3rd para p.13) is used but, as no doubt the authors are aware, there are multiple forms and methods to enable patients to provide feedback to the NHS (some online, some not); their study relates to one of these forms but this distinction is not always sufficiently clear. (The title of the manuscript is clear enough but in the abstract and on other occasions where the phrase 'patient feedback' is used further clarification would help). The opening sentence of the abstract states that the aim of the study is 'to examine the key areas of concern expressed ...'; this does not seem quite accurate as the study explores both positive and negative aspects of patient experiences. The authors should consider replacing 'concern' with phrasing which better describes their overall study. (this reoccurs on page 12 where the authors themselves go on to highlight that their study explores positive and negative feedback equally). Under the 'design' section of the abstract it would be helpful to include a second sentence summarising the analytical approach used. Under the 'participants' section of the abstract can the authors specify at what level contributors are posting comments, i.e. at a clinical service level, at an organisational level or at a system level. For readers unfamiliar with NHS Choices this will otherwise be unclear. Additionally, could the authors state how many contributors there were in the time period under study? In the 'conclusions' section of the abstract could the authors consider whether stating that technical competence is an 'important'
---

aspect of healthcare provision might be rephrased? 'Fundamental' surely?

In the 'conclusions' section of the abstract the authors state that their findings 'raise questions about the reliability of patient feedback exercises for evaluating the technical competence of practitioners'. Which exercises are they thinking of here? Which exercises focus solely on technical competence? I couldn't quite make the link between their findings and this statement.

In the 'Discussion' section of the manuscript (and the abstract) the authors highlight 'staff training' as a key recommendation; could they be more specific here? What form of training? (Quality Improvement?) At what stage of a healthcare professionals education? (Undergraduate? CPD)? How delivered? At the moment this is a rather generic statement and I think the authors should at least highlight that there are various mechanisms by which such 'training' might be offered/mandated? The first paragraph on p13 addresses these questions to some extent but are a little vague?

In the 'strengths and limitations of this study' section (the bullets on p2-3) the second bullet highlights that the study relates to a wider range of areas of health care provision. This rather tantalises the reader to think that comparisons will be included in the paper between the different forms of provision; in fact the authors do not present such comparisons but recommend this is an area for further work. I think this should be reflected in this bullet.

In the 'Discussion' section of the manuscript (and possibly the bulleted list at beginning) could the authors consider whether the fact that they have not studied and do not comment on whether the patient feedback was used in any way to improve services needs to be acknowledged? [S Ziebland, J Calabrese, A Coulter and L Locock (eds). 'Understanding and using experiences of health and illness', Oxford; Oxford University Press is a recent overview of the field and might be usefully drawn upon to broaden the discussion section. Ongoing research highly relevant to this paper also includes John Powell's INQUIRE study <http://www.nets.nihr.ac.uk/projects/hsdr/140448> & Carol Rivas's work <http://www.nets.nihr.ac.uk/projects/hsdr/1415615> it may be helpful to the authors to explore these study protocols and reflect on the future research agenda and implications for practice given these ongoing studies]

In the 'Methods' section of the manuscript I would like to see more detail as to the analytical process. Linguistic analysis and computer-assisted methods are not are area of my expertise but I wanted to know more about the process itself and other readers may feel the same. For example, how were the 10 most frequently 'linguistic markers' identified - what was the process? In the 'Results' section I was struck by the remark that it is the three words preceding and following the evaluation words that generated the lists - is this a standard approach in such analyses? As a non-expert I have no idea and would like to know whether this is an accepted approach or whether this study is employing novel methods? Importantly, how were the four areas derived? How were the data coded to arrive at these four areas?

Minor point - but I printed the manuscript in black and white and so figure 1 was unhelpful; I couldn't distinguish between negative and

	positive in the columns. A presentational issue - when the four areas are examined (p.8-10) might it be easier for the reader if all the % results were reported in an overall table rather than embedded in the text? Exemplar direct quotations from the postings relevant to each of the four areas and points being made would also help I think. Line 33, page 12 - typo/syntax? The ethics statement should include the full name of the ethics committee and reference number for the study.
--	---

VERSION 1 – AUTHOR RESPONSE

Review 1

Comment: The majority of the data comes from primary care and I wonder if the study would be more usefully conceptualised as a study of primary care rather than the NHS in general.

Response: Thanks for this suggestion. Indeed, the majority of the comments in our corpus relate to the primary care services of GP practices, hospitals and dentists. However, since the data also contains comments relating to things like care providers and mental health trusts, we did not re-conceptualise the study entirely. But we have included in the design section of the study an acknowledgement of this imbalance of the data, as well as acknowledging it in the ‘strengths and weaknesses’ section of the conclusion.

Comment: Please explain how you accessed the data from NHS Choices.

Response: We have now added more detail regarding how we extracted the data in the design section.

Comment: Please explain the process followed to identify the ten most frequently occurring linguistic markers of positive and negative evaluation.

Response: We have now added more information about this in the design section

Comment: Please explain how you sampled 100 messages for qualitative analysis. Given the size of your dataset, how did you ascertain that a qualitative examination of 100 comments would be sufficient? Did you aim for or reach saturation? Did you sample comments on all services?

Response: We have now included responses to all of these questions within our expanded and more detailed study design section.

Please explain the qualitative analytic process you followed to analyse the data, and supplement your qualitative findings with examples of the comments.

Response: We have now provided more detail regarding the manual reading and coding procedure we followed, which is also accompanied by example comments.

Comment: In the results section, table 2 shows the 10 most frequent positive and negative evaluative words. How can you be sure that the words were always used in a positive or negative way in the comments? For instance, the words good or great could have been negatively used as in 'not great' or 'not good'? Or the words bad or worst could have been used in comments as 'not bad' or 'not the worst'. I'm sure you've considered this, but please reflect on this in the paper.

Response: We have now acknowledged this in the analysis section and detailed our approach to this methodological obstacle.

Comment: One of the key findings is that comments frequently mentioned staff interpersonal skills. It would be helpful to know who is defined as 'staff' (this may well include GP receptionists, for instance) as this inevitably has implications for any recommendations based on your findings.

Response: When referring to staff the comments did indeed relate to medical and non-medical staff, such as receptionists, porters and cleaners. We have now clarified this at the earliest opportunity in the analysis section.

Comment: There is a growing body of research on online patient comments that has not been cited here. It is clear the authors are aware of this, but it would be helpful to position this study in relation to work conducted elsewhere.

Response: We have now expanded the amount of work cited substantially (increasing the number of references from 23 to 40). The literature on patient feedback (particularly online) is now discussed in the background section and referenced more extensively throughout the paper in its entirety.

Review 2

Comment: Throughout the manuscript it would be helpful if the authors could make clear to the reader that their study is focused on just one specific form of patient feedback (online comments and ratings) to one particular website (NHS Choices). In various places the phrase 'patient feedback' (e.g. first sentence, 3rd para p.13) is used but, as no doubt the authors are aware, there are multiple forms and methods to enable patients to provide feedback to the NHS (some online, some not); their study relates to one of these forms but this distinction is not always sufficiently clear. (The title of the manuscript is clear enough but in the abstract and on other occasions where the phrase 'patient feedback' is used further clarification would help).

Response: We have now made it clearer in the introduction section that patient feedback is given in numerous formats (online and offline) and that our study presents an analysis of the former (and from one particular online platform only). This point has also been worked into the conclusion and areas for future research.

Comment: The opening sentence of the abstract states that the aim of the study is 'to examine the key areas of concern expressed ...'; this does not seem quite accurate as the study explores both positive and negative aspects of patient experiences. The authors should consider replacing 'concern' with phrasing which better describes their overall study. (this reoccurs on page 12 where the authors themselves go on to highlight that their study explores positive and negative feedback equally).

Response: We have re-phrased all mentions of 'concerns' to reflect this broader conception, opting for 'drivers' of positive and negative feedback, where possible.

Comment: Under the 'design' section of the abstract it would be helpful to include a second sentence summarising the analytical approach used.

Response: We have added a second sentence providing more detail about our analytical procedure.

Comment: Under the 'participants' section of the abstract can the authors specify at what level contributors are posting comments, i.e. at a clinical service level, at an organisational level or at a system level. For readers unfamiliar with NHS Choices this will otherwise be unclear. Additionally, could the authors state how many contributors there were in the time period under study?

Response: We have now provided more information regarding how the feedback was given in the design section. For technical reasons it is not possible to ascertain exactly how many contributors are included in the data. However, we do know the minimum number, so have given that to at least provide an approximate figure.

Comment: In the 'conclusions' section of the abstract could the authors consider whether stating that technical competence is an 'important' aspect of healthcare provision might be rephrased? 'Fundamental' surely?

Response: We agree and have altered the wording to reflect this.

Comment: In the 'conclusions' section of the abstract the authors state that their findings 'raise questions about the reliability of patient feedback exercises for evaluating the technical competence of practitioners'. Which exercises are they thinking of here? Which exercises focus solely on technical competence? I couldn't quite make the link between their findings and this statement.

Response: Thank you for this observation. In the cold light of day this claim was problematic, so we have omitted it.

Comment: In the 'Discussion' section of the manuscript (and the abstract) the authors highlight 'staff training' as a key recommendation; could they be more specific here? What form of training? (Quality Improvement?) At what stage of a healthcare professionals education? (Undergraduate? CPD)? How delivered? At the moment this is a rather generic statement and I think the authors should at least highlight that there are various mechanisms by which such 'training' might be offered/mandated? The first paragraph on p13 addresses these questions to some extent but are a little vague?

Response: We have now expanded this recommendation regarding how and at what level these skills are perhaps best addressed.

Comment: In the 'strengths and limitations of this study' section (the bullets on p2-3) the second bullet highlights that the study relates to a wider range of areas of health care provision. This rather tantalises the reader to think that comparisons will be included in the paper between the different forms of provision; in fact the authors do not present such comparisons but recommend this is an area for further work. I think this should be reflected in this bullet.

Response: We have now pointed out in this bullet point that we do not make cross-area comparisons. However, the inclusion of comments relating to these areas does make for a more widely representative dataset.

Comment: In the 'Discussion' section of the manuscript (and possibly the bulleted list at beginning) could the authors consider whether the fact that they have not studied and do not comment on whether the patient feedback was used in any way to improve services needs to be acknowledged?

Response: The data we analyse has not yet been used to improve services. We have now made this clear in the discussion section.

Comment: Ongoing research highly relevant to this paper also includes John Powell's INQUIRE study <http://www.nets.nihr.ac.uk/projects/hsdr/140448> & Carol Rivas's work

<http://www.nets.nihr.ac.uk/projects/hsdr/1415615> it may be helpful to the authors to explore these study protocols and reflect on the future research agenda and implications for practice given these ongoing studies

Response: Thank you for these suggestions. We have found them very useful for expanding our coverage of the literature on patient feedback, which has in turn fed into our analysis and discussion.

Comment: In the 'Methods' section of the manuscript I would like to see more detail as to the analytical process. Linguistic analysis and computer-assisted methods are not an area of my expertise but I wanted to know more about the process itself and other readers may feel the same. For example, how were the 10 most frequently 'linguistic markers' identified - what was the process? In the 'Results' section I was struck by the remark that it is the three words preceding and following the evaluation words that generated the lists - is this a standard approach in such analyses? As a non-expert I have no idea and would like to know whether this is an accepted approach or whether this study is employing novel methods? Importantly, how were the four areas derived? How were the

data coded to arrive at these four areas?

Response: We have extensively revised and extended the design section so that it is more detailed regarding the specific procedures we used, including (hopefully) being more accessible with readers who are not so familiar with computer-assisted linguistic analysis. The collocation window of three is fairly standard in computational linguistic methodology, and we have pointed this out and supported it with a reference.

Comment: Minor point - but I printed the manuscript in black and white and so figure 1 was unhelpful; I couldn't distinguish between negative and positive in the columns.

Response: Thank you for this observation. We have now changed the colours to light and dark grey so that they contrast in black and white printing.

Comment: A presentational issue - when the four areas are examined (p.8-10) might it be easier for the reader if all the % results were reported in an overall table rather than embedded in the text? Exemplar direct quotations from the postings relevant to each of the four areas and points being made would also help I think.

Response: Following this suggestion we have now included a series of tables which display the percentages for each category, along with example comments.

Comment: Line 33, page 12 - typo/syntax?

Response: Corrected

Comment: The ethics statement should include the full name of the ethics committee and reference number for the study.

Response: We have included the name of the ethics committee (Lancaster University). However, there isn't a reference number available.

VERSION 2 – REVIEW

REVIEWER	Glenn Robert King's College London, England
REVIEW RETURNED	04-Jan-2017

GENERAL COMMENTS	Thank you for the opportunity to review this revised paper - I have no further comments or suggestions for the authors.
---